# Logarithmic Time Online Multiclass prediction

**Anna Choromanska**
Courant Institute of Mathematical Sciences
New York, NY, USA
achoroma@cims.nyu.edu

**John Langford**
Microsoft Research
New York, NY, USA
jcl@microsoft.com

## Abstract

We study the problem of multiclass classification with an extremely large number of classes ($k$), with the goal of obtaining train and test time complexity logarithmic in the number of classes. We develop top-down tree construction approaches for constructing logarithmic depth trees. On the theoretical front, we formulate a new objective function, which is optimized at each node of the tree and creates dynamic partitions of the data which are both pure (in terms of class labels) and balanced. We demonstrate that under favorable conditions, we can construct logarithmic depth trees that have leaves with low label entropy. However, the objective function at the nodes is challenging to optimize computationally. We address the empirical problem with a new online decision tree construction procedure. Experiments demonstrate that this online algorithm quickly achieves improvement in test error compared to more common logarithmic training time approaches, which makes it a plausible method in computationally constrained large-$k$ applications.

## 1   Introduction

The central problem of this paper is computational complexity in a setting where the number of classes $k$ for multiclass prediction is very large. Such problems occur in natural language (Which translation is best?), search (What result is best?), and detection (Who is that?) tasks. Almost all machine learning algorithms (with the exception of decision trees) have running times for multiclass classification which are $\mathcal{O}(k)$ with a canonical example being one-against-all classifiers [1].

In this setting, the most efficient possible accurate approach is given by information theory [2]. In essence, any multiclass classification algorithm must uniquely specify the bits of all labels that it predicts correctly on. Consequently, Kraft's inequality ([2] equation 5.6) implies that the expected *computational* complexity of predicting correctly is $\Omega(H(Y))$ per example where $H(Y)$ is the Shannon entropy of the label. For the worst case distribution on $k$ classes, this implies $\Omega(\log(k))$ computation is required.

Hence, our goal is achieving $O(\log(k))$ computational time per example[1] for both training and testing, while effectively using online learning algorithms to minimize passes over the data.

The goal of logarithmic (in $k$) complexity naturally motivates approaches that construct a logarithmic depth hierarchy over the labels, with one label per leaf. While this hierarchy is sometimes available through prior knowledge, in many scenarios it needs to be learned as well. This naturally leads to a *partition* problem which arises at each node in the hierarchy. The partition problem is finding a classifier: $c : X \rightarrow \{-1, 1\}$ which divides examples into two subsets with a purer set of labels than the original set. Definitions of purity vary, but canonical examples are the number of labels remaining in each subset, or softer notions such as the average Shannon entropy of the class labels. Despite resulting in a classifier, this problem is fundamentally different from standard binary classification. To see this, note that replacing $c(x)$ with $-c(x)$ is very bad for binary classification, but has no impact on the quality of a partition[2]. The partition problem is fundamentally non-convex

for symmetric classes since the average $\frac{c(x)-c(x)}{2}$ of $c(x)$ and $-c(x)$ is a poor partition (the always-0 function places all points on the same side).

The choice of partition matters in problem dependent ways. For example, consider examples on a line with label $i$ at position $i$ and threshold classifiers. In this case, trying to partition class labels $\{1, 3\}$ from class label 2 results in poor performance.

The partition problem is typically solved for decision tree learning via an enumerate-and-test approach amongst a small set of possible classifiers (see e.g. [3]). In the multiclass setting, it is desirable to achieve substantial error reduction for each node in the tree which motivates using a richer set of classifiers in the nodes to minimize the number of nodes, and thereby decrease the computational complexity. The main theoretical contribution of this work is to establish a boosting algorithm for learning trees with $O(k)$ nodes and $O(\log k)$ depth, thereby addressing the goal of logarithmic time train and test complexity. Our main theoretical result, presented in Section 2.3, generalizes a binary boosting-by-decision-tree theorem [4] to multiclass boosting. As in all boosting results, performance is critically dependent on the quality of the *weak learner*, supporting intuition that we need sufficiently rich partitioners at nodes. The approach uses a new objective for decision tree learning, which we optimize at each node of the tree. The objective and its theoretical properties are presented in Section 2.

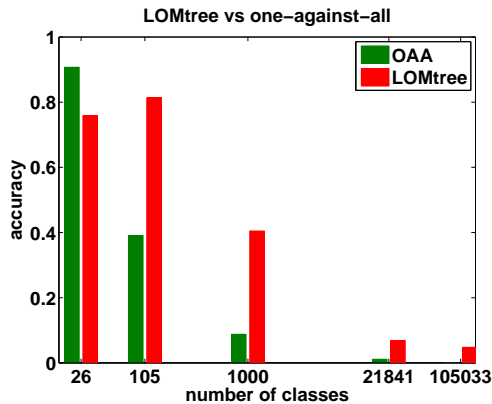

Figure 1: A comparison of One-Against-All (OAA) and the Logarithmic Online Multiclass Tree (LOMtree) with One-Against-All constrained to use the same training time as the LOMtree by dataset truncation and LOMtree constrained to use the same representation complexity as One-Against-All. As the number of class labels grows, the problem becomes harder and the LOMtree becomes more dominant.

A complete system with multiple partitions could be constructed top down (as the boosting theorem) or bottom up (as Filter tree [5]). A bottom up partition process appears impossible with representational constraints as shown in Section 6 in the Supplementary material so we focus on top-down tree creation.

Whenever there are representational constraints on partitions (such as linear classifiers), finding a strong partition function requires an efficient search over this set of classifiers. Efficient searches over large function classes are routinely performed via gradient descent techniques for supervised learning, so they seem like a natural candidate. In existing literature, examples for doing this exist when the problem is indeed binary, or when there is a prespecified hierarchy over the labels and we just need to find partitioners aligned with that hierarchy. Neither of these cases applies—we have multiple labels and want to dynamically create the choice of partition, rather than assuming that one was handed to us. Does there exist a purity criterion amenable to a gradient descent approach? The precise objective studied in theory fails this test due to its discrete nature, and even natural approximations are challenging to tractably optimize under computational constraints. As a result, we use the theoretical objective as a motivation and construct a new Logarithmic Online Multiclass Tree (LOMtree) algorithm for empirical evaluation.

Creating a tree in an online fashion creates a new class of problems. What if some node is initially created but eventually proves useless because no examples go to it? At best this results in a wasteful solution, while in practice it starves other parts of the tree which need representational complexity. To deal with this, we design an efficient process for recycling orphan nodes into locations where they are needed, and prove that the number of times a node is recycled is at most logarithmic in the number of examples. The algorithm is described in Section 3 and analyzed in Section 3.1.

And is it effective? Given the inherent non-convexity of the partition problem this is unavoidably an empirical question which we answer on a range of datasets varying from 26 to 105K classes in Section 4. We find that under constrained training times, this approach is quite effective compared to all baselines while dominating other $O(\log k)$ train time approaches.

What's new? To the best of our knowledge, the splitting criterion, the boosting statement, the LOMtree algorithm, the swapping guarantee, and the experimental results are all new here.

## 1.1 Prior Work

Only a few authors address logarithmic time training. The Filter tree [5] addresses consistent (and robust) multiclass classification, showing that it is possible in the statistical limit. The Filter tree does not address the partition problem as we do here which is shown in our experimental section is often helpful. The partition finding problem is addressed in the conditional probability tree [6], but that paper addresses conditional probability estimation. Conditional probability estimation can be converted into multiclass prediction [7], but doing so is not a logarithmic time operation.

Quite a few authors have addressed logarithmic testing time while allowing training time to be $O(k)$ or worse. While these approaches are intractable on our larger scale problems, we describe them here for context. The partition problem can be addressed by recursively applying spectral clustering on a confusion graph [8] (other clustering approaches include [9]). Empirically, this approach has been found to sometimes lead to badly imbalanced splits [10]. In the context of ranking, another approach uses $k$-means hierarchical clustering to recover the label sets for a given partition [11].

The more recent work [12] on the multiclass classification problem addresses it via sparse output coding by tuning high-cardinality multiclass categorization into a bit-by-bit decoding problem. The authors decouple the learning processes of coding matrix and bit predictors and use probabilistic decoding to decode the optimal class label. The authors however specify a class similarity which is $O(k^2)$ to compute (see Section 2.1.1 in [12]), and hence this approach is in a different complexity class than ours (this is also born out experimentally). The variant of the popular error correcting output code scheme for solving multi-label prediction problems with large output spaces under the assumption of output sparsity was also considered in [13]. Their approach in general requires $O(k)$ running time to decode since, in essence, the fit of each label to the predictions must be checked and there are $O(k)$ labels. Another approach [14] proposes iterative least-squares-style algorithms for multi-class (and multi-label) prediction with relatively large number of examples and data dimensions, and the work of [15] focusing in particular on the cost-sensitive multiclass classification. Both approaches however have $O(k)$ training time.

Decision trees are naturally structured to allow logarithmic time prediction. Traditional decision trees often have difficulties with a large number of classes because their splitting criteria are not well-suited to the large class setting. However, newer approaches [16, 17] have addressed this effectively at significant scales in the context of multilabel classification (multilabel learning, with missing labels, is also addressed in [18]). More specifically, the first work [16] performs brute force optimization of a multilabel variant of the Gini index defined over the set of positive labels in the node and assumes label independence during random forest construction. Their method makes fast predictions, however has high training costs [17]. The second work [17] optimizes a rank sensitive loss function (Discounted Cumulative Gain). Additionally, a well-known problem with hierarchical classification is that the performance significantly deteriorates lower in the hierarchy [19] which some authors solve by biasing the training distribution to reduce error propagation while simultaneously combining bottom-up and top-down approaches during training [20].

The reduction approach we use for optimizing partitions implicitly optimizes a differential objective. A non-reductive approach to this has been tried previously [21] on other objectives yielding good results in a different context.

## 2 Framework and theoretical analysis

In this section we describe the essential elements of the approach, and outline the theoretical properties of the resulting framework. We begin with high-level ideas.

### 2.1 Setting

We employ a hierarchical approach for learning a multiclass decision tree structure, training this structure in a *top-down* fashion. We assume that we receive examples $x \in \mathcal{X} \subseteq \mathbb{R}^d$, with labels $y \in \{1, 2, \ldots, k\}$. We also assume access to a hypothesis class $\mathcal{H}$ where each $h \in \mathcal{H}$ is a binary classifier, $h : \mathcal{X} \mapsto \{-1, 1\}$. The overall objective is to learn a tree of depth $O(\log k)$, where each node in the tree consists of a classifier from $\mathcal{H}$. The classifiers are trained in such a way that $h_n(x) = 1$ ($h_n$ denotes the classifier in node $n$ of the tree[3]) means that the example $x$ is sent to the right subtree of node $n$, while $h_n(x) = -1$ sends $x$ to the left subtree. When we reach a leaf, we predict according to the label with the highest frequency amongst the examples reaching that leaf.

In the interest of computational complexity, we want to encourage the number of examples going to the left and right to be *fairly balanced*. For good statistical accuracy, we want to send examples of class $i$ almost exclusively to either the left or the right subtree, thereby refining the *purity* of the class distributions at subsequent levels in the tree. The *purity* of a tree node is therefore a measure of whether the examples of each class reaching the node are then mostly sent to its one child node (pure split) or otherwise to both children (impure split). The formal definitions of *balancedness* and *purity* are introduced in Section 2.2. An objective expressing both criteria[4] and resulting theoretical properties are illustrated in the following sections. A key consideration in picking this objective is that we want to effectively optimize it over hypotheses $h \in \mathcal{H}$, while streaming over examples in an online fashion[5]. This seems unsuitable with some of the more standard decision tree objectives such as Shannon or Gini entropy, which leads us to design a new objective. At the same time, we show in Section 2.3 that under suitable assumptions, optimizing the objective also leads to effective reduction of the average Shannon entropy over the entire tree.

## 2.2 An objective and analysis of resulting partitions

We now define a criterion to measure the quality of a hypothesis $h \in \mathcal{H}$ in creating partitions at a fixed node $n$ in the tree. Let $\pi_i$ denotes the proportion of label $i$ amongst the examples reaching this node. Let $P(h(x) > 0)$ and $P(h(x) > 0|i)$ denote the fraction of examples reaching $n$ for which $h(x) > 0$, marginally and conditional on class $i$ respectively. Then we define the objective[6]:

$$J(h) = 2 \sum_{i=1}^{k} \pi_i \left| P(h(x) > 0) - P(h(x) > 0|i) \right|. \tag{1}$$

We aim to *maximize the objective* $J(h)$ to obtain high quality partitions. Intuitively, the objective encourages the fraction of examples going to the right from class $i$ to be substantially different from the background fraction for each class $i$. As a concrete simple scenario, if $P(h(x) > 0) = 0.5$ for some hypothesis $h$, then the objective prefers $P(h(x) > 0|i)$ to be as close to 0 or 1 as possible for each class $i$, leading to pure partitions. We now make these intuitions more formal.

**Definition 1** (Purity). *The hypothesis $h \in \mathcal{H}$ induces a pure split if*

$$\alpha := \sum_{i=1}^{k} \pi_i \min(P(h(x) > 0|i), P(h(x) < 0|i)) \leq \delta,$$

*where $\delta \in [0, 0.5)$, and $\alpha$ is called the* purity factor.

In particular, a partition is called *maximally pure* if $\alpha = 0$, meaning that each class is sent exclusively to the left or the right. We now define a similar definition for the balancedness of a split.

**Definition 2** (Balancedness). *The hypothesis $h \in \mathcal{H}$ induces a balanced split if*

$$c \leq \underbrace{P(h(x) > 0)}_{=\beta} \leq 1 - c,$$

*where $c \in (0, 0.5]$, and $\beta$ is called the* balancing factor.

A partition is called *maximally balanced* if $\beta = 0.5$, meaning that an equal number of examples are sent to the left and right children of the partition. The balancing factor and the purity factor are related as shown in Lemma 1 (the proofs of Lemma 1 and the following lemma (Lemma 2) are deferred to the Supplementary material).

**Lemma 1.** *For any hypothesis $h$, and any distribution over examples $(x, y)$, the purity factor $\alpha$ and the balancing factor $\beta$ satisfy $\alpha \leq \min\{(2 - J(h))/(4\beta) - \beta, 0.5\}$.*

A partition is called *maximally pure and balanced* if it satisfies both $\alpha = 0$ and $\beta = 0.5$. We see that $J(h) = 1$ for a hypothesis $h$ inducing a maximally pure and balanced partition as captured in the next lemma. Of course we do not expect to have hypotheses producing maximally pure and balanced splits in practice.

**Lemma 2.** *For any hypothesis $h : \mathcal{X} \mapsto \{-1, 1\}$, the objective $J(h)$ satisfies $J(h) \in [0, 1]$. Furthermore, if $h$ induces a maximally pure and balanced partition then $J(h) = 1$.*

## 2.3 Quality of the entire tree

The above section helps us understand the quality of an individual split produced by effectively maximizing $J(h)$. We next reason about the quality of the entire tree as we add more and more nodes. We measure the quality of trees using the average entropy over all the leaves in the tree, and track the decrease of this entropy as a function of the number of nodes. Our analysis extends the theoretical analysis in [4], originally developed to show the boosting properties of the decision trees for binary classification problems, to the multiclass classification setting.

Given a tree $\mathcal{T}$, we consider the entropy function $G_t$ as the measure of the quality of tree:

$$G_t = \sum_{l \in \mathcal{L}} w_l \sum_{i=1}^{k} \pi_{l,i} \ln \left( \frac{1}{\pi_{l,i}} \right)$$

where $\pi_{l,i}$'s are the probabilities that a randomly chosen data point $x$ drawn from $\mathcal{P}$, where $\mathcal{P}$ is a fixed target distribution over $\mathcal{X}$, has label $i$ given that $x$ reaches node $l$, $\mathcal{L}$ denotes the set of all tree leaves, $t$ denotes the number of internal tree nodes, and $w_l$ is the weight of leaf $l$ defined as the probability a randomly chosen $x$ drawn from $\mathcal{P}$ reaches leaf $l$ (note that $\sum_{l \in \mathcal{L}} w_l = 1$).

We next state the main theoretical result of this paper (it is captured in Theorem 1). We adopt the *weak learning* framework. The *weak hypothesis assumption*, captured in Definition 3, posits that each node of the tree $\mathcal{T}$ has a hypothesis $h$ in its hypothesis class $\mathcal{H}$ which guarantees simultaneously a "weak" purity and a "weak" balancedness of the split on any distribution $\mathcal{P}$ over $\mathcal{X}$. Under this assumption, one can use the new decision tree approach to drive the error below any threshold.

**Definition 3** (Weak Hypothesis Assumption). *Let $m$ denote any node of the tree $\mathcal{T}$, and let $\beta_m = P(h_m(x) > 0)$ and $P_{m,i} = P(h_m(x) > 0|i)$. Furthermore, let $\gamma \in \mathbb{R}^+$ be such that for all $m$, $\gamma \in (0, \min(\beta_m, 1 - \beta_m)]$. We say that the* weak hypothesis assumption *is satisfied when for any distribution $\mathcal{P}$ over $\mathcal{X}$ at each node $m$ of the tree $\mathcal{T}$ there exists a hypothesis $h_m \in \mathcal{H}$ such that $J(h_m)/2 = \sum_{i=1}^{k} \pi_{m,i} |P_{m,i} - \beta_m| \geq \gamma$.*

**Theorem 1.** *Under the Weak Hypothesis Assumption, for any $\alpha \in [0, 1]$, to obtain $G_t \leq \alpha$ it suffices to make $t \geq (1/\alpha)^{\frac{4(1-\gamma)^2 \ln k}{\gamma^2}}$ splits.*

We defer the proof of Theorem 1 to the Supplementary material and provide its sketch now. The analysis studies a tree construction algorithm where we recursively find the leaf node with the highest weight, and choose to split it into two children. Let $n$ be the heaviest leaf at time $t$. Consider splitting it to two children. The contribution of node $n$ to the tree entropy changes after it splits. This change (entropy reduction) corresponds to a gap in the Jensen's inequality applied to the concave function, and thus can further be lower-bounded (we use the fact that Shannon entropy is strongly concave with respect to $\ell_1$-norm (see e.g., Example 2.5 in Shalev-Shwartz [24])). The obtained lower-bound turns out to depend proportionally on $J(h_n)^2$. This implies that the larger the objective $J(h_n)$ is at time $t$, the larger the entropy reduction ends up being, which further reinforces intuitions to maximize $J$. In general, it might not be possible to find any hypothesis with a large enough objective $J(h_n)$ to guarantee sufficient progress at this point so we appeal to a *weak learning assumption*. This assumption can be used to further lower-bound the entropy reduction and prove Theorem 1.

## 3 The LOMtree Algorithm

The objective function of Section 2 has another convenient form which yields a simple online algorithm for tree construction and training. Note that Equation 1 can be written (details are shown in Section 12 in the Supplementary material) as

$$J(h) = 2\mathbb{E}_i[|\mathbb{E}_x[\mathbb{1}(h(x) > 0)] - \mathbb{E}_x[\mathbb{1}(h(x) > 0|i)]|].$$

Maximizing this objective is a discrete optimization problem that can be relaxed as follows

$$J(h) = 2\mathbb{E}_i[|\mathbb{E}_x[h(x)] - \mathbb{E}_x[h(x)|i]|],$$

where $E_x[h(x)|i]$ is the expected score of class i.

We next explain our empirical approach for maximizing the relaxed objective. The empirical estimates of the expectations can be easily stored and updated online in every tree node. The decision whether to send an example reaching a node to its left or right child node is based on the sign of the difference between the two expectations: $\mathbb{E}_x[h(x)]$ and $\mathbb{E}_x[h(x)|y]$, where $y$ is a label of the data point, i.e. when $\mathbb{E}_x[h(x)] - \mathbb{E}_x[h(x)|y] > 0$ the data point is sent to the left, else it is sent to the right. This procedure is conveniently demonstrated on a toy example in Section 13 in the Supplement.

During training, the algorithm assigns a unique label to each node of the tree which is currently a leaf. This is the label with the highest frequency amongst the examples reaching that leaf. While

---
**Algorithm 1** LOMtree algorithm (online tree training)
___

**Input**: regression algorithm $R$, max number of tree non-leaf nodes $T$, swap resistance $R_S$
___
Subroutine **SetNode** ($v$)
___
$\boldsymbol{m}_v = \emptyset$   ($\boldsymbol{m}_v(y)$ - sum of the scores for class $y$)
$\boldsymbol{l}_v \; = \emptyset$   ($\boldsymbol{l}_v(y)$ - number of points of class $y$ reaching $v$)
$\boldsymbol{n}_v \; = \emptyset$   ($\boldsymbol{n}_v(y)$ - number of points of class $y$ which are used to train regressor in $v$)
$\boldsymbol{e}_v \; = \emptyset$   ($\boldsymbol{e}_v(y)$ - expected score for class $y$)
$\boldsymbol{E}_v = 0$   (expected total score)
$C_v = 0$   (the size of the smallest leaf[7] in the subtree with root $v$)
___
Subroutine **UpdateC** ($v$)
___
**While** ($v \neq r$ AND $C_{\text{PARENT}(v)} \neq C_v$)
    $v = \text{PARENT}(v); \quad C_v = \min(C_{\text{LEFT}(v)}, C_{\text{RIGHT}(v)})$[8]
___
Subroutine **Swap** (v)
___
Find a leaf $s$ for which $(C_s = C_r)$
$s_{\text{PA}} = \text{PARENT}(s); \; s_{\text{GPA}} = \text{GRANDPA}(\text{s}); \; s_{\text{SIB}} = \text{SIBLING}(s)$[9]
**If** $(s_{\text{PA}} = \text{LEFT}(s_{\text{GPA}}))$ LEFT$(s_{\text{GPA}}) = s_{\text{SIB}}$    **Else** RIGHT$(s_{\text{GPA}}) = s_{\text{SIB}}$
**UpdateC** ($s_{\text{SIB}}$);   **SetNode** ($s$);   LEFT$(v) = s$;   **SetNode** ($s_{\text{PA}}$);   RIGHT$(v) = s_{\text{PA}}$
___
**Create** root $r = 0$: **SetNode** ($r$);   $t = 1$
**For each** example $(\boldsymbol{x}, y)$ **do**
    Set $j = r$
    **While** $j$ is not a leaf **do**
       **If** $(l_j(y) = \emptyset)$
          $m_j(y) = 0; \quad l_j(y) = 0; \quad n_j(y) = 0; \quad e_j(y) = 0$
       **If** $(E_j > \boldsymbol{e}_j(y))$ $c = -1$    **Else** $c = 1$
       **Train** $h_j$ with example $(\boldsymbol{x}, c)$: $R(\boldsymbol{x}, c)$
       $l_j(y)$++;   $\boldsymbol{n}_j(y)$ ++;   $\boldsymbol{m}_j(y)$ += $h_j(\boldsymbol{x})$;   $\boldsymbol{e}_j(y) = \boldsymbol{m}_j(y)/\boldsymbol{n}_j(y)$;   $E_j = \frac{\sum_{i=1}^{k} \boldsymbol{m}_j(i)}{\sum_{i=1}^{k} \boldsymbol{n}_j(i)}$[10]
       **Set** $j$ to the child of $j$ corresponding to $h_j$
    **If**($j$ is a leaf)
       $l_j(y)$++
       **If**($\boldsymbol{l}_j$ has at least 2 non-zero entries)
          **If**($t < T$ OR $C_j - \max_i \boldsymbol{l}_j(i) > R_S(C_r + 1)$)
             **If** ($t < T$)
                **SetNode** (LEFT$(j)$);   **SetNode** (RIGHT$(j)$);   $t$++
             **Else Swap**(j)
             $C_{\text{LEFT}(j)} = \lfloor C_j/2 \rfloor; \quad C_{\text{RIGHT}(j)} = C_j - C_{\text{LEFT}(j)}; \quad$ **UpdateC** (LEFT$(j)$)
      $C_j$++
___

testing, a test example is pushed down the tree along the path from the root to the leaf, where in each non-leaf node of the path its regressor directs the example either to the left or right child node. The test example is then labeled with the label assigned to the leaf that this example descended to.

The training algorithm is detailed in Algorithm 1 where each tree node contains a classifier (we use linear classifiers), i.e. $h_j$ is the regressor stored in node $j$ and $h_j(\mathbf{x})$ is the value of the prediction of $h_j$ on example $\mathbf{x}$[11]. The stopping criterion for expanding the tree is when the number of non-leaf nodes reaches a threshold $T$.

### 3.1 Swapping

Consider a scenario where the current training example descends to leaf $j$. The leaf can split (create two children) if the examples that reached it in the past were coming from at least two different classes. However, if the number of non-leaf nodes of the tree reaches threshold $T$, no more nodes can be expanded and thus $j$ cannot create children. Since the tree construction is done online, some nodes created at early stages of training may end up useless because no examples reach them later

___

[7]The smallest leaf is the one with the smallest total number of data points reaching it in the past.

[8]PARENT(v), LEFT(v) and RIGHT(v) denote resp. the parent, and the left and right child of node $v$.

[9]GRANDPA(v) and SIBLING(v) denote respectively the grandparent of node $v$ and the sibling of node $v$, i.e. the node which has the same parent as $v$.

[10]In the implementation both sums are stored as variables thus updating $E_v$ takes $\mathcal{O}(1)$ computations.

[11]We also refer to this prediction value as the 'score' in this section.

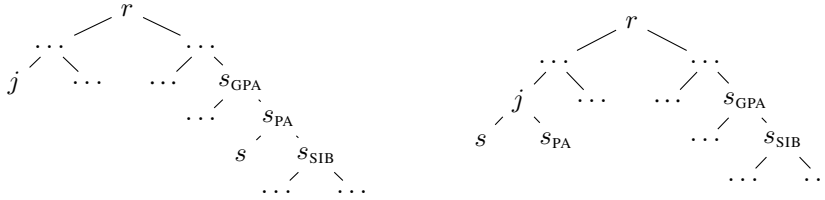

Figure 2: Illustration of the swapping procedure. **Left:** before the swap, **right:** after the swap.

on. This prevents potentially useful splits such as at leaf $j$. This problem can be solved by recycling orphan nodes (subroutine **Swap** in Algorithm 1). The general idea behind node recycling is to allow nodes to split if a certain condition is met. In particular, node $j$ splits if the following holds:

$$C_j - \max_{i \in \{1,2,\ldots,k\}} \boldsymbol{l}_j(i) > R_S(C_r + 1), \qquad (2)$$

where $r$ denotes the root of the entire tree, $C_j$ is the size of the smallest leaf in the subtree with root $j$, where the smallest leaf is the one with the smallest total number of data points reaching it in the past, $\boldsymbol{l}_j$ is a $k$-dimensional vector of non-negative integers where the $i^{\text{th}}$ element is the count of the number of data points with label $i$ reaching leaf $j$ in the past, and finally $R_S$ is a "swap resistance". The subtraction of $\max_{i \in \{1,2,\ldots,k\}} \boldsymbol{l}_j(i)$ in Equation 2 ensures that a pure node will not be recycled.

If the condition in Inequality 2 is satisfied, the swap of the nodes is performed where an orphan leaf $s$, which was reached by the smallest number of examples in the past, and its parent $s_{\text{PA}}$ are detached from the tree and become children of node $j$ whereas the old sibling $s_{\text{SIB}}$ of an orphan node $s$ becomes a direct child of the old grandparent $s_{\text{GPA}}$. The swapping procedure is shown in Figure 2. The condition captured in the Inequality 2 allows us to prove that the number of times any given node is recycled is upper-bounded by the logarithm of the number of examples whenever the swap resistance is $4$ or more (Lemma 3).

**Lemma 3.** *Let the swap resistance $R_S$ be greater or equal to $4$. Then for all sequences of examples, the number of times Algorithm 1 recycles any given node is upper-bounded by the logarithm (with base $2$) of the sequence length.*

## 4   Experiments

We address several hypotheses experimentally.

1. The LOMtree algorithm achieves true logarithmic time computation in practice.
2. The LOMtree algorithm is competitive with or better than all other logarithmic train/test time algorithms for multiclass classification.
3. The LOMtree algorithm has statistical performance close to more common $O(k)$ approaches.

Table 1: Dataset sizes.

| | Isolet | Sector | Aloi | ImNet | ODP |
|---|---|---|---|---|---|
| size | 52.3MB | 19MB | 17.7MB | 104GB[12] | 3GB |
| # features | 617 | 54K | 128 | 6144 | 0.5M |
| # examples | 7797 | 9619 | 108K | 14.2M | 1577418 |
| # classes | 26 | 105 | 1000 | ~22K | ~105K |

To address these hypotheses, we conducted experiments on a variety of benchmark multiclass datasets: *Isolet*, *Sector*, *Aloi*, *ImageNet* (*ImNet*) and *ODP*[13]. The details of the datasets are provided in Table 1. The datasets were divided into training ($90\%$) and testing ($10\%$). Furthermore, $10\%$ of the training dataset was used as a validation set.

The baselines we compared *LOMtree* with are a balanced random tree of logarithmic depth (*Rtree*) and the *Filter tree* [5]. Where computationally feasible, we also compared with a one-against-all classifier (*OAA*) as a representative $O(k)$ approach. All methods were implemented in the Vowpal Wabbit [25] learning system and have similar levels of optimization. The regressors in the tree nodes for *LOMtree*, *Rtree*, and *Filter tree* as well as the *OAA* regressors were trained by online gradient descent for which we explored step sizes chosen from the set $\{0.25, 0.5, 0.75, 1, 2, 4, 8\}$. We used linear regressors. For each method we investigated training with up to 20 passes through the data and we selected the best setting of the parameters (step size and number of passes) as the one minimizing the validation error. Additionally, for the *LOMtree* we investigated different settings of the stopping

criterion for the tree expansion: $T = \{k - 1, 2k - 1, 4k - 1, 8k - 1, 16k - 1, 32k - 1, 64k - 1\}$, and swap resistance $R_S = \{4, 8, 16, 32, 64, 128, 256\}$.

In Table 2 and 3 we report respectively train time and per-example test time (the best performer is indicated in bold). Training time (and later reported test error) is not provided for *OAA* on *ImageNet* and *ODP* due to intractability[14]-both are petabyte scale computations[15].

Table 2: Training time on selected problems.

|  | Isolet | Sector | Aloi |
|---|---|---|---|
| LOMtree | **16.27s** | **12.77s** | **51.86s** |
| OAA | 19.58s | 18.37s | 11m2.43s |

Table 3: Per-example test time on all problems.

|  | Isolet | Sector | Aloi | ImNet | ODP |
|---|---|---|---|---|---|
| LOMtree | **0.14ms** | **0.13ms** | **0.06ms** | **0.52ms** | **0.26ms** |
| OAA | 0.16 ms | 0.24ms | 0.33ms | 0.21s | 1.05s |

The first hypothesis is consistent with the experimental results. Time-wise *LOMtree* significantly outperforms *OAA* due to building only close-to logarithmic depth trees. The improvement in the training time increases with the number of classes in the classification problem. For instance on *Aloi* training with *LOMtree* is 12.8 times faster than with $OAA$. The same can be said about the test time, where the per-example test time for *Aloi*, *ImageNet* and *ODP* are respectively 5.5, 403.8 and 4038.5 times faster than *OAA*. The significant advantage of *LOMtree* over *OAA* is also captured in Figure 3.

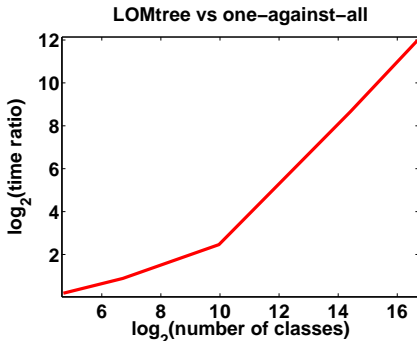

Figure 3: Logarithm of the ratio of per-example test times of $OAA$ and *LOMtree* on all problems.

Next, in Table 4 (the best logarithmic time performer is indicated in bold) we report test error of logarithmic train/test time algorithms. We also show the binomial symmetrical $95\%$ confidence intervals for our results. Clearly the second hypothesis is also consistent with the experimental results. Since the *Rtree* imposes a random label partition, the resulting error it obtains is generally worse than the error obtained by the competitor methods including *LOMtree* which learns the label partitioning directly from the data. At the same time *LOMtree* beats *Filter tree* on every dataset, though for *ImageNet* and *ODP* (both have a high level of noise) the advantage of *LOMtree* is not as significant.

Table 4: Test error (%) and confidence interval on all problems.

|  | LOMtree | Rtree | Filter tree | OAA |
|---|---|---|---|---|
| Isolet | **6.36**±1.71 | 16.92±2.63 | 15.10±2.51 | 3.56±1.30% |
| Sector | 16.19±2.33 | **15.77**±2.30 | 17.70±2.41 | 9.17±1.82% |
| Aloi | **16.50**±0.70 | 83.74±0.70 | 80.50±0.75 | 13.78±0.65% |
| ImNet | **90.17**±0.05 | 96.99±0.03 | 92.12±0.04 | NA |
| ODP | **93.46**±0.12 | 93.85±0.12 | 93.76±0.12 | NA |

The third hypothesis is weakly consistent with the empirical results. The time advantage of *LOMtree* comes with some loss of statistical accuracy with respect to *OAA* where *OAA* is tractable. We conclude that *LOMtree* significantly closes the gap between other logarithmic time methods and *OAA*, making it a plausible approach in computationally constrained large-$k$ applications.

## 5 Conclusion

The LOMtree algorithm reduces the multiclass problem to a set of binary problems organized in a tree structure where the partition in every tree node is done by optimizing a new partition criterion online. The criterion guarantees pure and balanced splits leading to logarithmic training and testing time for the tree classifier. We provide theoretical justification for our approach via a boosting statement and empirically evaluate it on multiple multiclass datasets. Empirically, we find that this is the best available logarithmic time approach for multiclass classification problems.

**Acknowledgments**

We would like to thank Alekh Agarwal, Dean Foster, Robert Schapire and Matus Telgarsky for valuable discussions.

## Footnotes

[1]Throughout the paper by logarithmic time we mean logarithmic time per example.

[2]The problem bears parallels to clustering in this regard.

[3]Further in the paper we skip index $n$ whenever it is clear from the context that we consider a fixed tree node.

[4]We want an objective to achieve its optimum for simultaneously pure and balanced split. The standard entropy-based criteria, such as Shannon or Gini entropy, as well as the criterion we will propose, posed in Equation 1, satisfy this requirement (for the entropy-based criteria see [4], for our criterion see Lemma 2).

[5]Our algorithm could also be implemented as batch or streaming, where in case of the latter one can for example make one pass through the data per every tree level, however for massive datasets making multiple passes through the data is computationally costly, further justifying the need for an online approach.

[6]The proposed objective function exhibits some similarities with the so-called Carnap's measure [22, 23] used in probability and inductive logic.

[12]compressed

[13]The details of the source of each dataset are provided in the Supplementary material.

[14]Note however that the mechanics of testing datastes are much easier - one can simply test with effectively untrained parameters on a few examples to measure the test speed thus the per-example test time for *OAA* on *ImageNet* and *ODP* is provided.

[15]Also to the best of our knowledge there exist no state-of-the-art results of the *OAA* performance on these datasets published in the literature.

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
