[Supplementary Material]

# Logarithmic Time Online Multiclass prediction (Supplementary Material)

## 6 Bottom-up partitions do not work

The most natural bottom-up construction for creating partitions is not viable as will be now shown by an example. Bottom-up construction techniques start by pairing labels, either randomly or arbitrarily, and then building a predictor of whether the class label is left or right conditioned on the class label being one of the paired labels. In order to construct a full tree, this operation must compose, pairing trees with size 2 to create trees of size 4. Here, we show that the straightforward approach to composition fails.

Suppose we have a one dimensional feature space with examples of class label $i$ having feature value $i$ and we work with threshold predictors. Suppose we have 4 classes $1, 2, 3, 4$, and we happen to pair $(1, 3)$ and $(2, 4)$. It is easy to build a linear predictor for each of these splits. The next step is building a predictor for $(1, 3)$ vs $(2, 4)$ which is impossible because all thresholds in $(-\infty, 1)$, $(2, 3)$, and $(4, \infty)$ err on two labels while thresholds on $(1, 2)$ and $(3, 4)$ err on one label.

## 7 Proof of Lemma 1

We start from deriving an upper-bound on $J(h)$. For the ease of notation let $P_i = P(h(x) > 0|i)$. Thus

$$J(h) = 2\sum_{i=1}^{k} \pi_i \left|P(h(x) > 0|i) - P(h(x) > 0)\right| = 2\sum_{i=1}^{k} \pi_i \left|P_i - \sum_{j=1}^{k} \pi_j P_j\right|,$$

where $\forall_{i=\{1,2,\ldots,k\}} 0 \le P_i \le 1$. Let $\alpha_i = \min(P_i, 1 - P_i)$ and recall the purity factor $\alpha = \sum_{i=1}^{k} \pi_i \alpha_i$ and the balancing factor $\beta = P(h(x) > 0)$. Without loss of generality let $\beta \le \frac{1}{2}$. Furthermore, let

$$L_1 = \{i : i \in \{1, 2, \ldots, k\}, P_i \ge \frac{1}{2}\}, \quad L_2 = \{i : i \in \{1, 2, \ldots, k\}, P_i \in [\beta, \frac{1}{2})\}$$

$$\text{and} \quad L_3 = \{i : i \in \{1, 2, \ldots, k\}, P_i < \beta\}.$$

First notice that

$$\beta = \sum_{i=1}^{k} \pi_i P_i = \sum_{i\in L_1} \pi_i(1 - \alpha_i) + \sum_{i \in L_2 \cup L_3} \pi_i \alpha_i = \sum_{i \in L_1} \pi_i - 2\sum_{i \in L_1} \pi_i \alpha_i + \alpha \qquad (3)$$

Therefore

$$
\begin{aligned}
\frac{J(h)}{2} &= \sum_{i=1}^{k} \pi_i |P_i - \beta| = \sum_{i \in L_1} \pi_i(1 - \alpha_i - \beta) + \sum_{i \in L_2} \pi_i(\alpha_i - \beta) + \sum_{i \in L_3} \pi_i(\beta - \alpha_i) \\
&= \sum_{i \in L_1} \pi_i(1 - \beta) - \sum_{i \in L_1} \pi_i \alpha_i + \sum_{i \in L_2} \pi_i \alpha_i - \sum_{i \in L_2} \pi_i \beta + \sum_{i \in L_3} \pi_i \beta - \sum_{i \in L_3} \pi_i \alpha_i
\end{aligned}
$$

Note that $\sum_{i \in L_3} \pi_i = 1 - \sum_{i \in L_1} \pi_i - \sum_{i \in L_2} \pi_i$ and therefore

$$
\begin{aligned}
\frac{J(h)}{2} &= \sum_{i \in L_1} \pi_i(1 - \beta) - \sum_{i \in L_1} \pi_i \alpha_i + \sum_{i \in L_2} \pi_i \alpha_i - \sum_{i \in L_2} \pi_i \beta + \beta(1 - \sum_{i \in L_1} \pi_i - \sum_{i \in L_2} \pi_i) - \sum_{i \in L_3} \pi_i \alpha_i \\
&= \sum_{i \in L_1} \pi_i(1 - 2\beta) - \sum_{i \in L_1} \pi_i \alpha_i + \sum_{i \in L_2} \pi_i \alpha_i + \beta(1 - 2\sum_{i \in L_2} \pi_i) - \sum_{i \in L_3} \pi_i \alpha_i
\end{aligned}
$$

Furthermore, since $-\sum_{i\in L_1}\pi_i\alpha_i + \sum_{i\in L_2}\pi_i\alpha_i - \sum_{i\in L_3}\pi_i\alpha_i = -\alpha + 2\sum_{i\in L_2}\pi_i\alpha_i$ we further write that

$$\frac{J(h)}{2} = \sum_{i\in L_1}\pi_i(1-2\beta) + \beta(1 - 2\sum_{i\in L_2}\pi_i) - \alpha + 2\sum_{i\in L_2}\pi_i\alpha_i$$

By Equation 3, it can be further rewritten as

$$\frac{J(h)}{2} = (1-2\beta)(\beta + 2\sum_{i\in L_1}\pi_i\alpha_i - \alpha) + \beta(1 - 2\sum_{i\in L_2}\pi_i) - \alpha + 2\sum_{i\in L_2}\pi_i\alpha_i$$

$$= 2(1-\beta)(\beta-\alpha) + 2(1-2\beta)\sum_{i\in L_1}\pi_i\alpha_i + 2\sum_{i\in L_2}\pi_i(\alpha_i - \beta)$$

Since $\alpha_i$'s are bounded by 0.5 we obtain

$$\frac{J(h)}{2} \leq 2(1-\beta)(\beta-\alpha) + 2(1-2\beta)\sum_{i\in L_1}\pi_i\alpha_i + 2\sum_{i\in L_2}\pi_i(\frac{1}{2} - \beta)$$

$$\leq 2(1-\beta)(\beta-\alpha) + 2(1-2\beta)\alpha + 1 - 2\beta$$
$$= 2\beta(1-\beta) - 2\alpha(1-\beta) + 2\alpha(1-2\beta) + 1 - 2\beta$$
$$= 1 - 2\beta^2 - 2\beta\alpha$$

Thus:

$$\alpha \leq \frac{2 - J(h)}{4\beta} - \beta.$$

## 8 Proof of Lemma 2

*Proof.* We first show that $J(h) \in [0,1]$. We start from deriving an upper-bound on $J(h)$, where $h \in \mathcal{H}$ is some hypothesis in the hypothesis class. For the ease of notation let $P_i = P(h(x) > 0|i)$. Thus

$$J(h) = 2\sum_{i=1}^{k}\pi_i |P(h(x) > 0|i) - P(h(x) > 0)| \qquad (4)$$

$$= 2\sum_{i=1}^{k}\pi_i \left| P_i - \sum_{j=1}^{k}\pi_j P_j \right|,$$

where $\forall_{i=\{1,2,...,k\}} 0 \leq P_i \leq 1$. The objective $J(h)$ is certainly maximized on the extremes of the $[0,1]$ interval. The upper-bound on $J(h)$ can be thus obtained by setting some of the $P_i$'s to 1's and remaining ones to 0's. To be more precise, let

$$L_1 = \{i : i \in \{1,2,\ldots,k\}, P_i = 1\} \quad \text{and} \quad L_2 = \{i : i \in \{1,2,\ldots,k\}, P_i = 0\}.$$

Therefore it follows that

$$J(h) \leq 2\left[\sum_{i\in L_1}\pi_i(1 - \sum_{j\in L_1}\pi_j) + \sum_{i\in L_2}\pi_i\sum_{j\in L_1}\pi_j\right]$$

$$= 2\left[\sum_{i\in L_1}\pi_i - (\sum_{i\in L_1}\pi_i)^2 + (1 - \sum_{i\in L_1}\pi_i)\sum_{i\in L_1}\pi_i\right]$$

$$= 4\left[\sum_{i\in L_1}\pi_i - (\sum_{i\in L_1}\pi_i)^2\right]$$

Let $b = \sum_{i\in L_1}\pi_i$ thus

$$J(h) \leq 4b(1-b) = -4b^2 + 4b \qquad (5)$$

Since $b \in [0,1]$, it is straightforward that $-4b^2 + 4b \in [0,1]$ and thus $J(h) \in [0,1]$.

We now proceed to prove the main statement of Lemma 2, if $h$ induces a maximally pure and balanced partition then $J(h) = 1$. Since $h$ is maximally balanced, $P(h(x) > 0) = 0.5$. Simultaneously, since $h$ is maximally pure $\forall_{i=\{1,2,...,k\}}(P(h(x) > 0|i) = 0 \text{ or } P(h(x) > 0|i) = 1)$. Substituting that into Equation 5 yields that $J(h) = 1$. $\square$

# 9 Proof of Theorem 1

*Proof.* The analysis studies a tree construction algorithm where we recursively find the leaf node with the highest weight, and choose to split it into two children. Consider the tree constructed over $t$ steps where in each step we take one leaf node and split it into two. Let $n$ be the heaviest node at time $t$ and its weight $w_n$ be denoted by $w$ for brevity. Consider splitting this leaf to two children $n_0$ and $n_1$. For the ease of notation let $w_0 = w_{n_0}$ and $w_1 = w_{n_1}$. Also for the ease of notation let $\beta = P(h_n(x) > 0)$ and $P_i = P(h_n(x) > 0|i)$. Let $\pi_i$ be the shorthand for $\pi_{n,i}$ and $h$ be the shorthand for $h_n$. Recall that $\beta = \sum_{i=1}^k \pi_i P_i$ and $\sum_{i=1}^k \pi_i = 1$. Also notice that $w_0 = w(1-\beta)$ and $w_1 = w\beta$. Let $\boldsymbol{\pi}$ be the $k$-element vector with $i^{th}$ entry equal to $\pi_i$. Furthermore let $\tilde{G}(\boldsymbol{\pi}) = \sum_{i=1}^k \pi_i \ln\left(\frac{1}{\pi_i}\right)$.

Before the split the contribution of node $n$ to $G_t$ was $w\tilde{G}(\boldsymbol{\pi})$. Let $\pi_{n_0,i} = \frac{\pi_i(1-P_i)}{1-\beta}$ and $\pi_{n_1,i} = \frac{\pi_i P_i}{\beta}$ be the probabilities that a randomly chosen $x$ drawn from $\mathcal{P}$ has label $i$ given that $x$ reaches nodes $n_0$ and $n_1$ respectively. For brevity, let $\pi_{n_0,i}$ be denoted by $\pi_{0,i}$ and $\pi_{n_1,i}$ be denoted by $\pi_{1,i}$. Furthermore let $\boldsymbol{\pi}_0$ be the $k$-element vector with $i^{th}$ entry equal to $\pi_{0,i}$ and let $\boldsymbol{\pi}_1$ be the $k$-element vector with $i^{th}$ entry equal to $\pi_{1,i}$. Notice that $\boldsymbol{\pi} = (1-\beta)\boldsymbol{\pi}_0 + \beta\boldsymbol{\pi}_1$. After the split the contribution of the same, now internal, node $n$ changes to $w((1-\beta)\tilde{G}(\boldsymbol{\pi}_0) + \beta\tilde{G}(\boldsymbol{\pi}_1))$. We denote the difference between them as $\Delta_t$ and thus

$$\Delta_t := G_t - G_{t+1} = w\left[\tilde{G}(\boldsymbol{\pi}) - (1-\beta)\tilde{G}(\boldsymbol{\pi}_0) - \beta\tilde{G}(\boldsymbol{\pi}_1)\right]. \tag{6}$$

We aim to lower-bound $\Delta_t$. The entropy reduction of Equation 6 [4] corresponds to a gap in the Jensen's inequality applied to the concave function $\tilde{G}(\boldsymbol{\pi})$. This leads to the lower-bound on $\Delta_t$ given in Lemma 4 (the lemma is proven in Section 10 in the Supplementary material).

**Lemma 4.** *The entropy reduction $\Delta_t$ of Equation 6 can be lower-bounded as follows*

$$\Delta_t \geq \frac{J(h)^2 G_t}{8\beta(1-\beta)t\ln k}$$

Lemma 4 implies that the larger the objective $J(h)$ is at time $t$, the larger the entropy reduction ends up being, which further reinforces intuitions to maximize $J$. In general, it might not be possible to find any hypothesis with a large enough objective $J(h)$ to guarantee sufficient progress at this point so we appeal to a *weak learning assumption*. This assumption can be used to further lower-bound $\Delta_t$. The lower-bound can then be used (details are in Section 9 in the Supplementary material) to obtain the main theoretical statement of the paper captured in Theorem 1.

From the definition of $\gamma$ it follows that $1 - \gamma \geq \beta \geq \gamma$. Also note that the *weak hypothesis assumption* guarantees $J(h) \geq 2\gamma$, which applied to the lower-bound on $\Delta_t$ captured in Lemma 4 yields

$$\Delta_t \geq \frac{\gamma^2 G_t}{2(1-\gamma)^2 t\ln k}.$$

Let $\eta = \sqrt{\frac{8}{(1-\gamma)^2\ln k}}\gamma$. Then $\Delta_t > \frac{\eta^2 G_t}{16t}$. Thus we obtain the recurrence inequality

$$G_{t+1} \leq G_t - \Delta_t < G_t - \frac{\eta^2 G_t}{16t} = G_t\left[1 - \frac{\eta^2}{16t}\right]$$

One can now compute the minimum number of splits required to reduce $G_t$ below $\alpha$, where $\alpha \in [0,1]$. Applying the proof technique from [4] (the proof of Theorem 10) gives the final statement of Theorem 1. $\square$

# 10 Proof of Lemma 4

*Proof.* Without loss of generality assume that $P_1 \leq P_2 \leq \cdots \leq P_k$. As mentioned before, the entropy reduction $\Delta_t$ corresponds to a gap in the Jensen's inequality applied to the concave function $\tilde{G}(\boldsymbol{\pi})$. Also recall that Shannon entropy is strongly concave with respect to $\ell_1$-norm (see e.g., Example 2.5 in Shalev-Shwartz [24]). As a specific consequence (see e.g. Theorem 2.1.9 in Nesterov [26]) we obtain

$$\Delta_t \geq w\beta(1-\beta)\|\boldsymbol{\pi}_0 - \boldsymbol{\pi}_1\|_1^2 = \frac{w}{\beta(1-\beta)}\left(\sum_{i=1}^k |\pi_i(P_i - \beta)|\right)^2 = \frac{wJ(h)^2}{4\beta(1-\beta)}, \tag{7}$$

where the last equality results from the definition of $J(h) = 2 \sum_{i=1}^{k} \pi_i |P_i - \beta|$.

Note that the following holds $w \geq \frac{G_t}{2t \ln k}$, where recall that $w$ is the weight of the heaviest leaf in the tree, i.e. the leaf with the highest weight, at round $t$. This leaf is selected to the currently considered split [4]. In particular, the lower-bound on $w$ is the consequence of the following

$$G_t = \sum_{l \in \mathcal{L}} w_l \sum_{i=1}^{k} \pi_{l,i} \ln\left(\frac{1}{\pi_{l,i}}\right) \leq \sum_{l \in \mathcal{L}} w_l \ln k \leq 2tw \ln k,$$

where $w = \max_{l \in \mathcal{L}} w_l$. Thus $w \geq \frac{G_t}{2t \ln k}$ which when substituted to Equation 7 gives the final statement of the lemma. $\qquad \square$

## 11 Proof of Lemma 3

*Proof.* We bound the number of swaps that any node makes. Consider $R_S = 4$ and let $j$ be the node that is about to split and $s$ be the orphan node that will be recycled (thus $C_r = C_s$). The condition in Equation 2 implies that the swap is done if $C_j > 4(C_r + 1) = 4(C_s + 1)$. Algorithm 1 makes $s$ a child of $j$ during the swap and sets its counter to $C_s^{new} = \lfloor C_j/2 \rfloor \geq 2(C_r + 1) = 2(C_s + 1)$. Then $C_r$ gets updated. Since the value of $C_s^{new}$ at least doubles after a swap and all counters are bounded by the number of examples $n$, the node can be involved in at most $\log_2 n$ swaps. $\qquad \square$

## 12 Equivalent forms of the objective function

Consider the objective function as given in Equation 1

$$J(h) = 2 \sum_{i=1}^{k} \pi_i \left| P(h(x) > 0) - P(h(x) > 0|i) \right|.$$

Recall that $\mathcal{X}$ denotes the set of all examples and let $\mathcal{X}_i$ denote the set of examples in class $i$. Also let $|\mathcal{X}|$ denote the cardinality of set $\mathcal{X}$ and let $|\mathcal{X}_i|$ denote the cardinality of set $\mathcal{X}_i$. Then we can re-write the objective as

$$
\begin{aligned}
J(h) &= 2 \sum_{i=1}^{k} \pi_i \left| \frac{\sum_{x \in \mathcal{X}} \mathbb{1}(h(x) > 0)}{|\mathcal{X}|} - \frac{\sum_{x \in \mathcal{X}_i} \mathbb{1}(h(x) > 0)}{|\mathcal{X}_i|} \right| \\
&= 2 \sum_{i=1}^{k} \pi_i \left| \mathbb{E}_x[\mathbb{1}(h(x) > 0)] - \mathbb{E}_x[\mathbb{1}(h(x) > 0|i)] \right| \\
&= 2 \mathbb{E}_i[|\mathbb{E}_x[\mathbb{1}(h(x) > 0)] - \mathbb{E}_x[\mathbb{1}(h(x) > 0|i)]|].
\end{aligned}
$$

# 13 Toy example of the behavior of LOMtree algorithm

Figure 4 shows the toy example of the behavior of LOMtree algorithm for the first few data points. Without loss of generality we consider the root node (exactly the same actions would be performed in any other tree node). Notice that the algorithm achieves simultaneously balanced and pure split of classes reaching the considered node.

$e$ denotes the expectation $\mathbb{E}_x[h(x)]$, and $e1, e2, e3, e4$ denote the expectations $\mathbb{E}_x[h(x)|i = 1]$, $\mathbb{E}_x[h(x)|i = 2]$, $\mathbb{E}_x[h(x)|i = 3]$, and $\mathbb{E}_x[h(x)|i = 4]$. For simplicity we assume score $h(x)$ can only be either $1$ (if the example is sent to the right) or $-1$ (if the example is sent to the left). The figure should be read as follows (we explain how to read first few illustrations):

a) Root is initialized. Expectation $e$ is initialized to $0$.

b) The first example $x1$ comes with label $1$ (we denote it as $(x1, 1)$). $e1$ is initialized to $0$. The difference between $e$ and $e1$ is computed: $e - e1 = 0$. The difference is non-positive thus the example is sent to the right child of the root, which is now being created (the left child is created along with the right child as we always create both children of any node simultaneously).

c) Expectations $e$ and $e1$ get updated. It is shown that root and its right child saw an example of class $1$.

d) The second example $x2$ comes with label $2$ (we denote it as $(x2, 2)$). $e2$ is initialized to $0$. The difference between $e$ and $e2$ is computed: $e - e2 = 1$. The difference is positive thus the example is sent to the left child of the root.

e) Expectations $e$ and $e2$ get updated. It is shown that root saw examples of class $1$ and $2$, whereas its resp. left and right child saw example of class resp. $2$ and $1$.

f) ...

Figure 4: Toy example of the behavior of LOMtree algorithm in the tree root.

## 14 Experiments - dataset details

Below we provide the details of the datasets that we were using for the experiments in Section 4:

- *Isolet*: downloaded from `http://www.cs.huji.ac.il/˜shais/datasets/ClassificationDatasets.html`
- *Sector* and *Aloi*: downloaded from `http://www.csie.ntu.edu.tw/˜cjlin/libsvmtools/datasets/multiclass.html`
- *ImageNet* [27]: features extracted according to `http://www.di.ens.fr/willow/research/cnn/`, dataset obtained from the authors.
- *ODP* [20]: obtained from Paul Bennett. Our version has significantly more classes than reported in the cited paper because we use the entire dataset.