[Reviews · NeurIPS 2015]

Submitted by Assigned_Reviewer_1

Heavy review

Summary:

The paper concerns multi-class problems with a large number of classes. It introduces a novel label tree classifier that learns and predicts in logarithmic time in the number of classes. Theoretical guarantees in terms of a boosting-like theorem have been proven. Moreover, not only node classifiers, but also the structure of the tree is trained online. Additionally, the authors show a simple subtree swapping procedure that ensures proper balancing of the tree. Empirical results show attractiveness of the introduced approach.

Quality:

The paper presents both theoretical and empirical results. The method is clearly derived and theoretically justified. Some minor comments are given below.

The objective after reformulation can be stated as:

J(h) = 2 \sum_{i=1}^k | \Pr(i) \Pr(h(x) > 0) - \Pr(i, h(x) > 0) |

This is a kind of an independence test between i and h(x) > 0. I suppose that such measures have been used in rule induction (e.g., in generation of association rules) and decision tree learning. This is also similar to the so-called Carnap's confirmation measure (see, "Comparison of confirmation measures", Cognition, 2007, and "Logical foundations of probability (2nd ed.).", 1962).

Decision tree learning can be introduced as a recurrent procedure of empirical risk minimization guided by a surrogate loss function, where both splits and leaf predictions are computed as empirical risk minimizers (the notion of a purity measure is not needed then). For C4.5, this is logistic loss, and for CART squared error loss. It would be interesting to check whether there exists a loss function that corresponds to J(h).

The authors do not clearly describe the difference between LOMTree and standard decision trees. Footnote 4 states that standard criteria such as Shannon or Gini entropy satisfy requirement of balanced and pure splits. Since it is possible to adapt standard decision trees to the problems with a large number of classes [16,17], it is not clear what is the difference between these two approaches. Can we substitute J(h) by Shannon entropy?

I suppose that there exists a kind of a trade-off between test-time complexity and the complexity of the function class, i.e., by minimizing the number of prediction steps we should probably extend the complexity of the function class. This would explain the empirical results of OAA and LOMTree. Could you comment on this?

The prediction time results are given per test example. The overall method is also designed for improving test time for a single test example. However, when test examples are coming in mini-batches different tricks can be used to improve test time. How to use this method in such scenario?

Clarity:

The paper is clearly written.

Originality:

The paper is very original as it clearly states the problem of efficient learning and prediction in case of large output spaces. The introduced algorithm is also novel, however, the authors should take into account that the problem of decision tree induction was exhaustively studied in 80' and 90'. Many algorithms have already been introduced that share similarities with LOMtrees (incremental learning of trees, trees for data streams, Hoeffding trees, trees with linear splits).

Significance:

The paper will be (already is) very influential for large-scale machine learning.

After rebuttal:

Thank you for your responses.

I gave the references to confirmation measures since this was the most similar measure I knew to your objective. It would be great if you will explore this topic in the extended (e.g., arxiv or journal) version of the article. I have found your objective very interesting, but I think that similar measures have already been considered in many settings and many interesting links can be found. I agree that such measures are not so popular and successful in decision tree learning, especially in the multi-class setting.

The same concerns incremental learning of decision trees. There are some approaches that you can easily find in Internet. The question is how they relate to your approach. One well-know approach are Very Fast Decision Trees (aka Hoeffding trees) introduced by Domingos and Hulten (2000). A detailed discussion can be postponed however to the extended version of your paper.

Summary: This is a very inspiring paper that will be (or even already is) very influential for large scale machine learning. It should be accepted for publication at NIPS.

Submitted by Assigned_Reviewer_2

A nice work proposing to learn logarithmic online multiclass trees for efficient multiclass prediction. The authors introduce the principle behind LOMTree and its advantage wrt OAA classifiers. Then they propose a simple objective value for finding a partition on a given node that will promote maximally pure and balanced partition. Then the global quality of the tree is expressed as an entropy function. The proposed algorithm converges provably to a tree with log(k) complexity. Numerical experiments are performed on several datasets from 26 to 105k classes and show reasonable training and testing time.

This is a good paper well written but pretty dense. The proposed approach is novel and compares favorably to other tree-based classifiers while being close to OAA performances. I just have a few comments in the remaining.

- First the example for the construction of a node in the supplementary material is particularly clear. It would be nice in the paper instead of supplementary material (in parallel to algorithm?).

- The linear classifiers h at each nodes are estimated online. Is the update a stochastic gradient wrt J()? A few more detail would go a long way to reproducible research.

- The numerical experiments are a bit short. It would be interesting to add in table 4 the performance of OAA classification to help for the comparison (for the small datasets at least). They are given in the text but should appear in the table.

Summary: Interesting paper with a novel algorithm for constructing an efficient tree based classifier. Theoretical study and numerical experiments are convincing.

Submitted by Assigned_Reviewer_3

This paper proposes a novel online algorithm for constructing a multiclass classifier that enjoys a time complexity logarithmic in the number of classes k. This is done by constructing online a decision tree which locally maximizes an appropriate novel objective function, which measures the quality of a tree according to a combined "balancedness" and "purity" score. A theoretical analysis (of a probably intractable algorithm) is provided via a boosting argument (assuming weak learnability), essentially extending the work of Kearns and Mansour (1996) to the multiclass setup. A concrete algorithm is given to a relaxed problem (but see below) without any guarantees, but quite simple, natural and interesting.

The paper is well written, very interesting and seems to be original. The theoretical analysis seems to be correct. The experimental results suggest a good improvement over other logarithmic time classifiers but fall quite short of obtaining state-of-the-art generalization performance. I see this work as a small step forward toward this goal.

I do have some concerns:

- Obviously, the 1-nearest neighbour classifier has train and test times which are independent of the number of classes k. This is so since with the time scale used in your time complexity analysis, reading log(k) bits is regarded as a O(1) operation. Thus I find your claim that the most efficient possible accurate approach has time complexity log(k) as misleading. I don't find the information theoretic arguments given in the introduction as relevant to your time complexity analysis of the algorithm. Can you please clarify?

- If I understand correctly, the LOMtree algorithm does not quite maximize the relaxed objective given in line 263.

This is due to the fact that your estimates of the expectations m_v(y), E_v, etc. are not updated when some retrained hypothesis up the tree changes its decision on some samples along the paths. As the statistics given in algorithm 1 are additive, such changes does not reflect in your estimates. Isn't a more complex mechanism (such as message passing) is needed for accurate bookkeeping? It seems that such "open-loop" inaccurate statistics in the nodes can make the algorithm behave quite erratically. Can you please clarify this point?

- It seems that when one goes down in the tree, the classifiers are doomed to have lower accuracy. This is so since the tree is quite balanced and lower nodes will have much less samples to train a good classifier on. Thus, I suspect that an exponential number of samples will be needed to get good generalization accuracy.
Summary: This paper proposes an online multiclass classifier with sufficient novelty and interest to accept for publication.

Author Feedback
Author rebuttal: We thank the Reviewers for their valuable feedback. We incorporated all specific comments into our current draft and respond below.

Rev1:
-We added reference to Tentori et al 2007 and Carnap 1962. Note that Carnap's confirmation measure was never successfully applied to decision trees, and especially in the multi-class setting. Our objective also differs from the formulation of Carnap's measure in Tentori et al 2007, Section 1. Please mention other related citations we missed regarding incremental learning etc. so we can add them.

The reformulation of the objective is nicely symmetric.

-Standard decision tree splitting criteria are nontrivial to optimize with a classifier because they are based on the (discrete) split of the data. [21] shows how to optimize Shannon or Gini entropy via softening the split with a link function and heavy regularization. Essentially, substituting alternate J(h) via this method seems possible but provides no obvious additional desirable properties and may make things worse.

[16] performs brute force optimization of a multilabel variant of the Gini index defined over the set of positive labels in the node and assumes label independence during random forest construction. Their method has high training costs (it is emphasized in [17]). [17] optimizes a multilabel rank sensitive loss function (Discounted Cumulative Gain). We have now added this explanation to the paper.

A mathematical relation linking Shannon entropy and our objective is captured in Lemma 4 (Supplement).

-LOMTree and OAA have similar representational complexities in terms of number of parameters. There may be tradeoffs in general, but we do not yet have a proof that a tradeoff is necessary.
-The algorithm can be easily extended to the mini-batch setting. The regressors in tree nodes as well as all expectations are then updated using mini-batch of examples.

Rev2:
-Space is very tight, but we'll examine moving the example.
-Yes, J(h) is optimized via a variant of stochastic gradient. Note that all of our code is public, and hence experiments should be much more reproducible than is typical.
Our objective motivates the use of absolute value based loss, while existing regret analysis for one-against-all motivates the use of squared loss (or more generally proper loss functions) there.
-The results of OAA are now added to Table 4.

Rev3:
-We don't agree with the claim about 1-NN classifier requiring constant time. For arbitrary metric spaces, the test-time complexity for 1-NN is Omega(d*N) per example where d is the number of dimensions and N is the number of training examples. This is so, because in order to find the nearest neighbor you must test proximity to every point in the training set (there are special cases where this time complexity can be reduced, but they require strong additional assumptions). Typically N>k where k is the number of labels, so 1-NN is exponentially slower than the O(d*log k) per example which we observe empirically and expect theoretically.
-We use a running average instead of the expectation for (obvious) computational reasons. When the learning rate decays to zero, the running average converges to the expectation of the relaxed objective.
-The lower nodes in the tree see much less samples, but at the same time they have much simpler problems (they separate less classes) and thus are much easier to train than the nodes higher in the tree.

Rev4:
-Doing additional experiments against [11,17] don't appear to address our hypotheses. In particular [11,17] are irrelevant to hypotheses 1 and 3 at the beginning of the experimental section and the algorithms are not obviously logarithmic time (for hypothesis 2) due to the iterative nature of subproblems they solve. To be clear, we believe these are good and interesting algorithms, but our goal here is a theoretically sound basis for logarithmic time prediction which works well in practice. A different interesting topic for a paper is a broad comparison of many different sublinear time algorithms.
-Each node needs to store the statistics of classes that reached this node. Therefore, the best-case memory consumption for these statistics is O(k*log k) and the worst-case is O(k^2).
-It is true that LOMTree uses O(kd) space. The problem of storage is an important problem, but not one that we attempt to address. Note that our current algorithm has similar memory storage as common baseline approaches, like OAA.

Rev6:
-The objective function is in-depth discussed in Section 2, which includes the intuitive interpretation. O(log k) training and test time are demonstrated in the experiments (also Theorem 1 has only logarithmic dependence on k and implies this running time when the objective can be relatively well-optimized in tree nodes).
-The suggested baselines are not useful for our comparisons since they are O(k) or worse in training, thus intractable by our standards for large k.